# Energy-efficient extraction of linear alkanes from various isomers using structured metal-organic framework membrane

Yuecheng Wang [1,2], Yujie Ban [1,2] ✉, Ziyi Hu[1] & Weishen Yang [1,2] ✉

Extraction of low concentration linear alkanes ($C_5$-$C_7$) from various isomers is critical for the petrochemical industry. At present, the separation of alkane isomers is mainly accomplished by distillation, which results in substantial energy expenditure. Metal-organic frameworks (MOFs) with well-tailored nanopores have been demonstrated to be capable of realizing molecule-level separation. In this study, oriented HKUST-1 membranes are formulated according to the morphology-biased principle and finally realized with a low dose synthesis method for terminating undesired crystal nucleation and growth. The fully exposed triangular sieving pore array of the membrane induces configuration entropic diffusion to split linear alkanes from mono-branched and di-branched isomers as well as their cyclical counterparts. Typically, the current separation technique consumes 91% less energy than vacuum distillation. Furthermore, our membranes can realize one-step extraction of *normal*-pentane, *normal*-hexane and *normal*-heptane from a ten-component alkane isomer solution that mimics light naphtha.

Separation of light normal-/iso-alkanes ($C_5$-$C_7$) (Table 1) is a pivotal process for the petrochemical industry—linear alkanes are to be cracked for the production of ethylene and propylene whereas branched isomers are to be used as gasoline ingredients due to their higher research octane numbers (RON)[1-3]. In addition, another challenge must be confronted when implementing oil refining-isomerization processes. It is imperative to efficiently extract low concentrations of linear alkane residues (-10%)[4-7], which are to be returned to catalytic reactors for cycling. Distillation is a dominant method for separating linear alkanes from various isomers, which relies on phase changes and has to endure enormous energy consumption and high installation costs. Membrane-based separation, as a result of the low energy penalty involved, has proven to be a potential alternative to distillation. However, the minor differences in the underlying chemistries and polarizabilities between linear alkanes and their isomers make molecule-level separation challenging. The size/shape of these molecules are deemed to be a handle available for their discrimination. One approach is to take advantage of well-tailored molecule-level nanopores to engender selectivity.

Metal-organic frameworks (MOFs) with well-organized porosity exhibit impressive performance in diversified applications including gas storage[8-12], catalysis[13,14], sensing[15-17], and especially molecular discrimination and separation[18-23]. A survey of the literature indicates that most studies have focused on achieving MOFs as adsorbents for the separation of alkane isomers. An arguably additional challenge that limits progress is that MOF adsorbents need to be desorbed and activated under high-temperature or vacuum conditions[24,25]. Molecular sieve MOF membranes are comparatively more desired because they are able to immediately split the mixture into two streams (permeant and retentate). Furthermore, if MOFs are used as membranes, the anisotropy of the porous architecture has the potential to add a new dimension to chemical separation—it is expectant to control the membrane orientation[26-30], that is, orderly arrangement of the specifically discriminative pores in a convenient way during synthesis.

HKUST-1 is an archetypal MOF with the combination of di-nuclear copper paddlewheel units and 1,3,5-benzenetricarboxylic acid (BTCA) to form a face-centered cubic network[31]. The BET surface area of the HKUST-1 powder is 1244 $m^2\,g^{-1}$ according to the experimental $N_2$

[1]State Key Laboratory of Catalysis, Dalian Institute of Chemical Physics, Chinese Academy of Sciences, 457 Zhongshan Road, 116023 Dalian, P. R. China. [2]University of Chinese Academy of Sciences, 19A Yuquan Road, 100049 Beijing, P. R. China. ✉e-mail: yjban@dicp.ac.cn; yangws@dicp.ac.cn

**Table 1 | Typical C₅-C₇ alkane structures and nomenclature**

| Name | Structure | Abbreviation |
|---|---|---|
| *normal*-Pentane | | *n*Pen |
| 2-Methylbutane | | 2MB |
| *normal*-Hexane | | *n*Hex |
| 2-Methylpentane | | 2MP |
| 3-Methylpentane | | 3MP |
| 2,2-Dimethylbutane | | 22DMB |
| 2,3-Dimethylbutane | | 23DMB |
| *cyclo*-Hexane | | cHex |
| *normal*-Heptane | | *n*Hep |
| 2-Methylhexane | | 2MH |
| 2,3-Dimethylpentane | | 23DMP |

adsorption isotherm at 77 K (Supplementary Fig. 1). The anisotropy of its porous architecture can be articulated when pore windows running through different directions are highlighted (Fig. 1a–c). Given that the rigid triangular window with a size of ~0.46 nm[32] is positioned between the kinetic diameters of alkane isomers, we speculate that {111}-oriented membranes with triangular-pore arrays are capable of splitting linear alkanes from branched (or cyclical counterparts) by virtue of the size repulsion mechanism.

Oriented membranes are highly reliant on MOF building blocks that can be organized on the surface of the support in an anisotropic manner. The core is to judiciously manipulate the morphology of building blocks and pack them closely at the interface. For HKUST-1, octahedron-shaped building blocks with highly exposed {111} crystal facets show a bias toward this orientation relative to cube- or truncated cuboctahedron (or cuboctahedron)-shaped building blocks[32,33], as shown in Fig. 1d, e. The previous literature also indicates that slower growth of the {111} crystal facets than {100} crystal facets generally makes the morphology of HKUST-1 crystals near a sharp octahedron that is dominated by {111} crystal facets. In our study, we find it possible to make the anisotropic growth more obvious at the support interface with a sharper vertical gradient in the concentration of nutrients, which can be well proven by the morphology difference between crystals growing at the interface (sharp octahedron) and at the bulk liquid phase (truncated cuboctahedron), as shown in Supplementary Fig. 2. With this in mind, for the synthesis of highly oriented HKUST-1 membranes composed of octahedron-shaped HKUST-1 building blocks, a vital prerequisite is to confine crystal nucleation and growth at the support interface rather than relying on crystal migration from the bulk solution to the support surface, and to deliberately isolate crystal nucleation and growth for independent control. Given that HKUST-1 is commonly shaped by slow nucleation and fast growth under ambient conditions, the temperature is thought to be a switcher available for dictating which process is dominant. Considering that crystal nucleation and growth at the support interface are commonly overwhelmed by these two behaviors in the bulk solution, this competition needs to be terminated to ensure a highly compact and

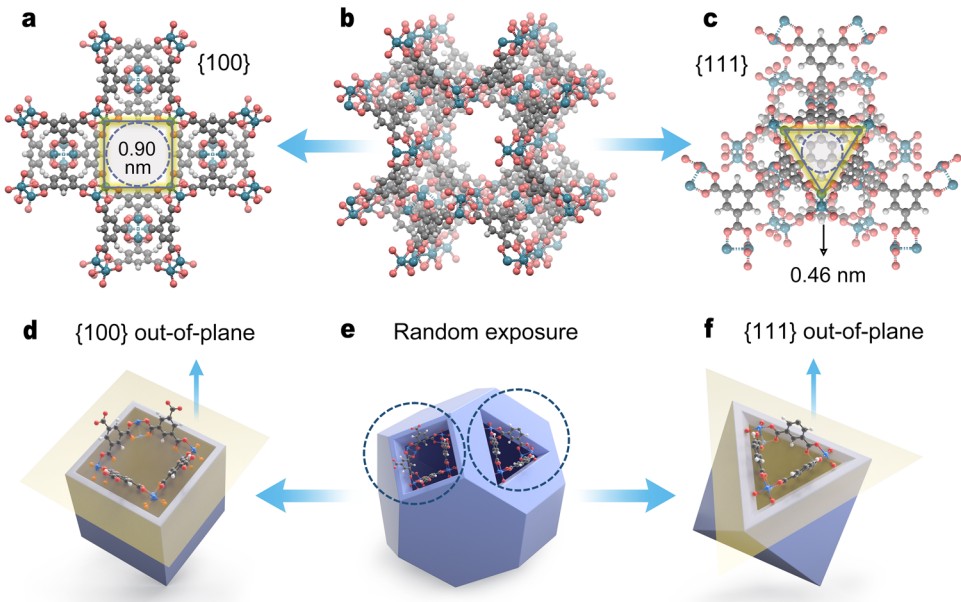

**Fig. 1 | The anisotropy of the porous HKUST-1 architecture. a–c** Specific pore windows running through the {100} and {111} crystal facets. The structure of HKUST-1 is shown in ball-and-stick models with Cu in blue, C in dark gray, O in red and H in light gray. Arrows indicate different view directions. **d, e** Orientation/ crystal facet arrangements with respect to the morphology-biased principle. The truncated cuboctahedron in **e** represents the equilibrium morphology of HKUST-1 according to the Bravais–Friedel–Donnay–Harker (BFDH) morphology simulation principle. Arrows point from **e** to **d** and **f** means the morphology evolution process.

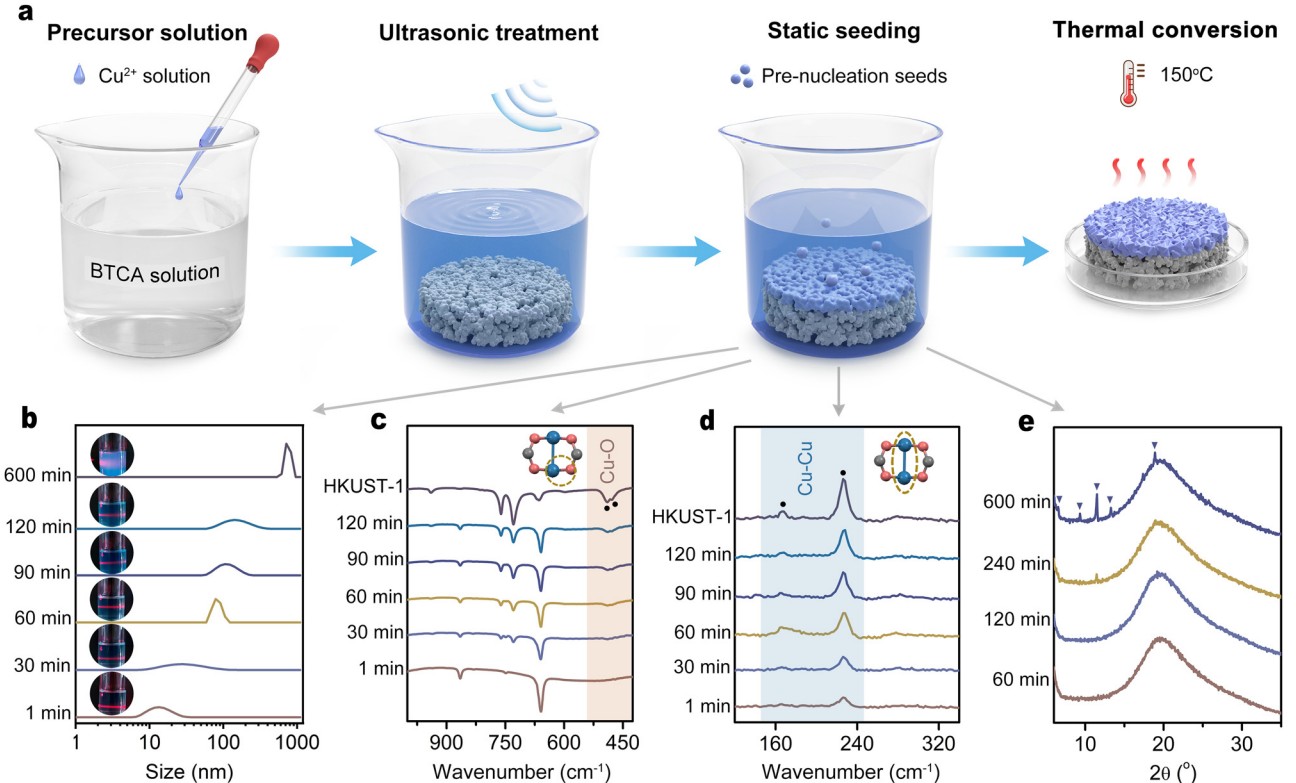

**Fig. 2 | The static seeding-thermal conversion (SS-TC) process for the synthesis of HKUST-1 membranes and attributes of seeds in the precursor solution.** **a** Schematic diagram for SS-TC with arrows indicating this process proceeding.

**b**–**e** Particle size distribution (**b**), FTIR spectra (**c**), Raman spectra (**d**), and XRD patterns (**e**) of seeds in the precursor solution (following arrows) plotted against static seeding times. Source data are provided as a Source data file.

oriented membrane. In light of the above analysis, we proposed a two-step strategy to synthesize HKUST-1 membranes with a {111} orientation (Fig. 2a): (1) gentle crystal nucleation in the form of the metal-linker pre-organization at the solution-alumina interface at room temperature to fulfil static seeding on the surface of the alumina support, and (2) thermal conversion by removing the bulk solution to trigger seed growth in a self-limiting chemical environment that lacks nutrients. The triangular-pore arrayed HKUST-1 membranes, as a result of the {111} orientation, can take advantage of entropic leverage for size/shape discrimination of linear alkanes from various isomers.

## Results and discussion
### Static seeding of pre-organized metal-linker species
The porous γ-phase alumina support was horizontally immersed into a clear precursor solution composed of copper (II) nitrate and BTCA, and treated in an ultrasonic bath for 2 min (Fig. 2a). Ultrasonication can help overcome the limitation of the triple-phase boundary involving alumina, air (in the pores of alumina) and liquid (Supplementary Fig. 3) and entrap nutrients in the pores of alumina for follow-up nucleation at the interface. After that, we kept the alumina support in the precursor solution to implement the static seeding process. Because it is difficult to directly detect seeds at the alumina−liquid boundary, we probe the precursor solution to obtain seed attributes.

During 1–120 min, there is a slight turbid phase arising in the precursor solution (Fig. 2b), which, however, cannot be collected by high-speed centrifugation, suggesting that metal-linker pre-organization followed by nucleation takes place in this solution under static conditions. After 1 min, Tyndall scattering is detected, and the central size of colloidal particles in the precursor solution is measured to be ca. 14.3 nm through a dynamic light scattering (DLS) instrument (Fig. 2b). For an extended time (typically 30–120 min), Tyndall scattering becomes attenuated, and the colloidal particles grow larger.

Notably, a relatively sharp profile centered at approximately 96.4 nm for the distribution of the colloidal particle size appears at 60 min, which becomes wide and blunt again with time. After 600 min, Tyndall scattering becomes weaker, concomitant with particles centered at ca. 753 nm beginning to precipitate out. Fourier transform infrared (FTIR) spectra of the colloidal solution confirm the appearance of the Cu-O vibration at 478 cm$^{-1}$ and 490 cm$^{-1}$ after 30 min at the earliest, which becomes pronounced with time (Fig. 2c). The Raman spectra further show Cu-Cu vibrations at 166 cm$^{-1}$ and 226 cm$^{-1}$ (Fig. 2d). Therefore, the structural chemistry of colloidal particles in the precursor solution can be described as BTC-bridged di-nuclear paddle-wheel species[34,35]. X-ray diffraction (XRD) was further used to record the crystallinity of colloidal particles in the solution in situ. Until 600 min, no adequate crystalline particles contribute to XRD (Fig. 2e). However, they are potential seeds to induce membrane formation and heavily influence the continuity and compactness of membranes, which is demonstrated by the surface morphology of membranes after 1 static seeding-thermal conversion (SS-TC) cycle and discussed hereafter. A time limit of 600 min was set for the static seeding process because there would be large particles precipitating from the bulk solution beyond this limit.

In addition, we also tried to directly characterize the pre-nucleation seed layer on the alumina support by scanning electron microscopy (SEM). Except for scattered crystals, we cannot obtain any surface morphology information of the pre-nucleation seeds because (1) these pre-organized metal-linker species on the top surface of the support might be susceptible to the high-powered electron beam and (2) they might penetrate into the alumina disk. We conducted element analysis along the cross-section of the disk by using an energy dispersive spectrometer (EDS) to probe the distribution of pre-nucleation seeds. A clear copper distribution reflecting pre-nucleation seeds along the cross-section of the disk is shown in Supplementary Fig. 4.

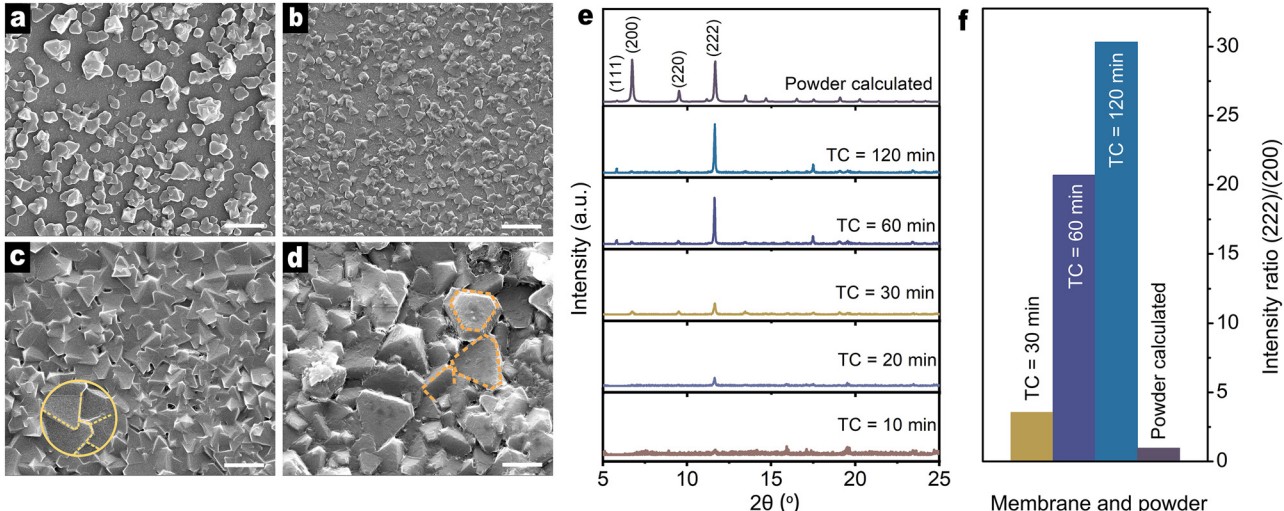

**Fig. 3 | Morphology and orientation of HKUST-1 membranes after 1 SS-TC cycle.**
**a**–**d** Top-view SEM images of membranes when the static seeding time is set for 1 min (**a**), 30 min (**b**), 60 min (**c**) and 120 min (**d**). The scale bar in **a**–**d** is 2 μm. **e** XRD patterns of membranes with different thermal conversion (TC) times. **f** The intensity ratio (222)/(200) of the membranes determined by experimental patterns and the standard powder calculated based on the structure CCDC 755080. Source data are provided as a Source data file.

Furthermore, the penetration depth of pre-nucleation seeds slightly changed with the seeding time, ranging from 0.60 to 0.82 μm for seeding times ranging from 30 to 120 min.

### Thermal conversion by removing the bulk solution

The static seeding process at the alumina-solution boundary was followed by thermal conversion (Fig. 2a). The alumina support was first removed from the precursor solution. Because of seeding, together with the capillary adsorption of nutrients on the surface of the support, the weight of the alumina disk increased by 21%. Before the thermal conversion process was initiated, the alumina support that carried the metal-organic pre-nucleation seeds and limited nutrients was placed in an open vessel, and transferred into an oven preheated at 150 °C. Typically, after 120 min, an SS-TC cycle was completed. SEM was used to probe the surface morphology of the membranes, which also reflects the sweet spot for the static seeding process (Fig. 3a–d). We notice that static seeding for 1–30 min is far from ideal because of the low packing density of small seeds. Therefore, the membranes after 1 cycle of SS-TC are discontinuous (Fig. 3a, b). For an extended seeding time, typically 60 min, the membrane surface after 1 cycle of SS-TC is completely covered by sharp octahedral crystals, with well-balanced triangular facets highly exposed out-of-plane (Fig. 3c). With a slight tilt, these octahedron crystals, similar to building blocks are tightly connected by edge fusion. From the surface morphology of the membrane, we can speculate that static seeding for 60 min hits the spot. In this case, ca. 96.4 nm of pre-nucleation seeds might be closely packed on the surface of the support in the static seeding process, which induces membrane densification after 1 cycle of SS-TC. The apparent thickness of the membrane, determined by the cross-sectional SEM image is ~0.6 μm (Supplementary Fig. 5). In addition, there is an approximately 0.3-μm-thick layer confined into the alumina support, which is proven by the penetration of copper into the Al-enrichment layer. Upon further prolonging static seeding (600 min), the crystalline seeds of ca. 753 nm lead to large quasi-octahedron crystals, by which with dislocation between, the membrane is imperfectly bricked up (Fig. 3d).

We also carried out two groups of comparative studies. In Group 1, the alumina support was immersed into a fresh precursor solution in a stainless-steel autoclave and immediately transferred into an oven to implement a solvothermal reaction. After that, we can only achieve scattered octahedron crystals on the surface of the support (Supplementary Fig. 6a), indicating the necessity of splitting nucleation (static seeding) from crystal growth (thermal conversion) by temperature in the membrane preparation process. In Group 2, the alumina support was kept in the bulk solution after static seeding, sealed in a stainless-steel autoclave and transferred into an oven. The deposition density of octahedral crystals slightly improves but still fails to result in a continuous membrane (Supplementary Fig. 6b). This finding is rationalized by considering the overwhelming advantage of the crystal growth in the bulk solution, which severely depletes nutrients and lessens the possibility of the seed growth at the alumina-solution interface. This reflects the significance of the thermal conversion process by eliminating the bulk solution.

### Orientation of HKUST-1 membranes

Time-variable thermal conversion trials were also conducted to trace the evolution of crystal facets in membranes. XRD shows a weak (222) diffraction peak pertaining to HKUST-1 crystals after thermal conversion for 20 min, yet with the absence of the (200) peak as a result of the limitation of the detection (Fig. 3e). For an extended time, for example, 30–120 min, we observe qualitatively that (222) diffractions become increasingly dominant. To quantify the orientation of the membranes, the intensity ratio (222)/(200) was employed. This value of membranes increases considerably with time, which indicates the gradual development of {111} crystal facets out-of-plane by the use of nutrients confined into the microvoids of the support. The membrane surface SEM images demonstrate the increasingly large crystals and their well-defined facets with time (Supplementary Fig. 7). We focus on the intensity ratio (222)/(200) of the final membrane after thermal conversion for 120 min, which roughly increases by a factor of 10 relative to the value of the membrane after thermal conversion for 30 min, and far exceeds the value determined based on the calculated powder diffractions (Fig. 3f). This comparative study suggests the dominant arrangement of {111} crystal facets out-of-plane, which matches well with the inspection of highly exposed triangular facets on the surface of the support through SEM (Fig. 3c).

Here we also discuss the orientation evolution mechanism with thermal conversion time. The thermal conversion process by removing the bulk solution is thought to be a process in which the crystal growth was completely confined at the support interface. The instantaneous

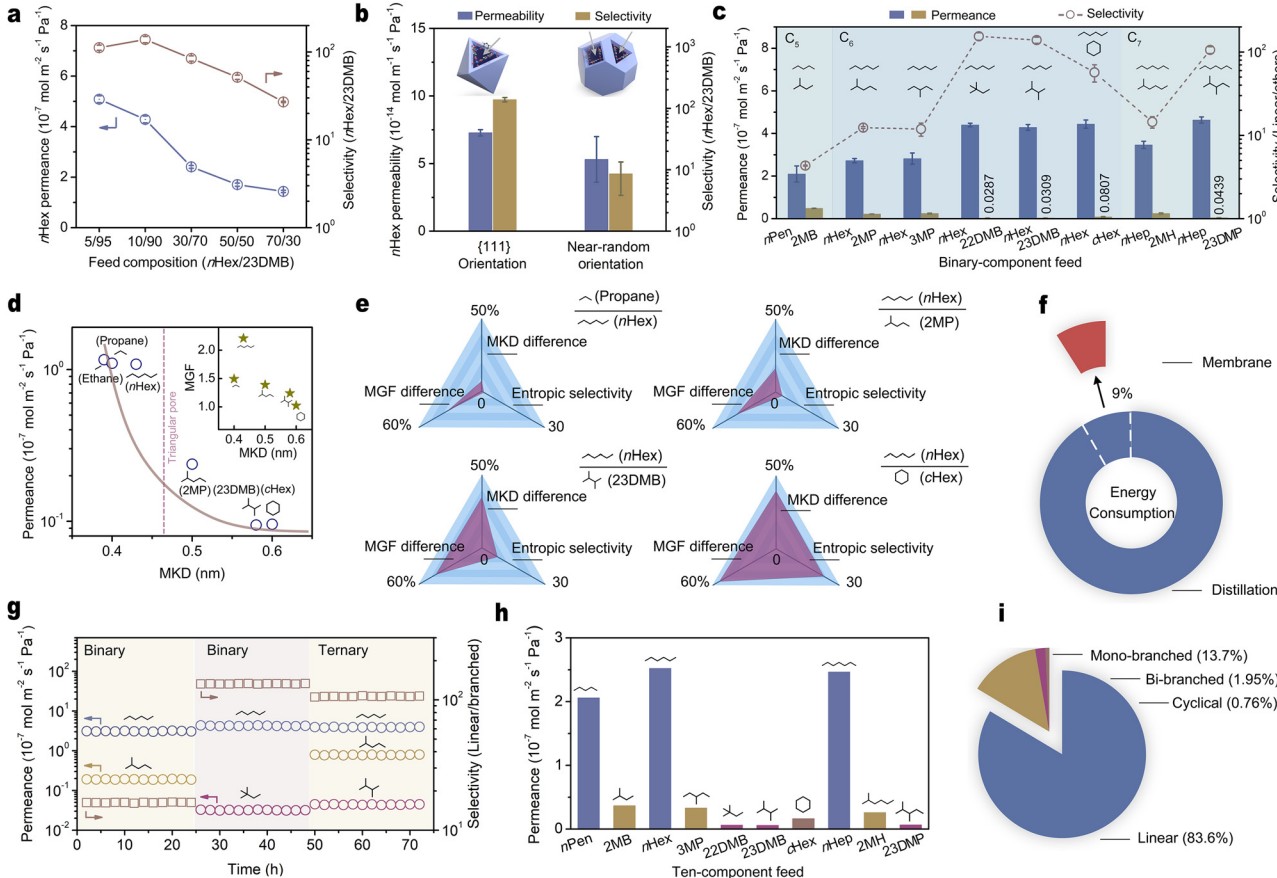

**Fig. 4 | C₅-C₇ alkane isomer separation through {111}-oriented HKUST−1 membranes at 30 °C. a** Separation properties toward different weight compositions of *n*Hex/23DMB feed solutions. **b** Comparison of separation properties between the {111} orientation and near-random orientation for the 10/90 *n*Hex/23DMB feed. **c** Separation properties toward various binary alkane isomers with the linear component accounting for 10 wt.%. **d** Single-component permeation of alkanes spanning a range of MKDs. **e** Diffusion entropic selectivity for alkane isomers with respect to their relative MKD and MGF differences. **f** Comparison of energy

consumption between membrane pervaporation and distillation. **g** A three-stage separation program with the introduction of 10/90 *n*Hex/2MP, 10/90 *n*Hex/23DMB and 10/10/80 *n*Hex/3MP/23DMB. **h** Separation properties toward the ten-component feed solution that mimics light naphtha. Each component accounts for 10 wt.% in the feed. **i** The weight spectrum of various alkanes in the permeant. Error bars in **a–c** represent the standard deviation of the results from three membranes. Source data are provided as a Source data file.

high-temperature environment can prompt the fast growth of seeds on the surface of the support by means of limited nutrients in the support. This self-limiting chemical condition at the support interface can widen the difference in the growth rate between {100} and {111} crystal facets, leading to a membrane composed of octahedral crystals. With the nutrients gradually depleted, the growth rate of {111} crystal facets could be further slowed down, thus leading to an increase in the proportion of triangular {111} facets aligned out-of-plane with time.

In addition to 150 °C, thermal conversion at other temperatures was also undertaken. A brief conclusion is that 150 °C, near the boiling point of DMF, is optimal. Thermal conversion at lower temperatures (typically 90 and 120 °C) produces smaller crystals with ill-defined facets, whereas thermal conversion at higher temperatures (typically 180 °C) causes a large number of holes on the surface of crystals because of fast evaporation of DMF (Supplementary Fig. 8). Overall, the membrane recovered from thermal conversion at 150 °C for 120 min immediately after static seeding for 60 min stands out by virtue of its desirable microscopic morphology. To eliminate minor local defects in the membranes completely and improve reproductivity, 2 cycles of SS-TC were adopted, by which the thickness of the membrane was increased to 1.7 μm whereas the intensity ratio (222)/(200) was barely changed (Supplementary Figs. 9 and 10), concurrent with a significant improvement in the isomer separation accuracy reaching 595% (Supplementary Fig. 11).

It is worth noting that a similar microscopic morphology, orientation and isomer separation performance can be achieved in the membrane if the thermal conversion process is sealed in a closed system (Supplementary Fig. 12). The thermal conversion process in the closed or open vessel has a negligible effect on the resultant membrane.

The {111}-oriented HKUST-1 membranes can also be prepared on titania supports, showing satisfactory isomer separation performances (Supplementary Figs. 13 and 14).

## Extraction of C₅-C₇ linear alkanes from isomers

Pervaporation through membranes for the separation of C₅-C₇ linear alkanes from branched isomers was carried out at 30 °C. Similar studies based on MOF membranes in the literature are rarely reported and remain a challenge. The HKUST−1 membranes in our study show separation potential for a wide composition of *n*Hex/23DMB (Fig. 4a). The membranes exhibit both significant *n*Hex permeance and *n*Hex/23DMB selectivity, beyond the recently reported Al-bttotb membrane[36]. Impressively, for the 10/90 *n*Hex/23DMB feed solution, the average selectivity of *n*Hex/23DMB through membranes is 139, equivalent to the concentrate of *n*Hex from 10 wt.% (in feed) to 91 wt.% (in permeant), which has important practical implications for integrated cyclic isomerization and process intensification[37]. The HKUST−1 membrane derived from the SS-TC method is closely connected to the support

similar to the MOF membrane synthesized by direct reaction of the alumina support with a linker[38,39]. After eight ultrasonic treatments (30 min each time), the permeance of *n*Hex and selectivity of *n*Hex/23DMB for the 10/90 *n*Hex/23DMB feed solution maintain constant (Supplementary Fig. 15). As expected, triangular-pore arrayed HKUST-1 membranes as a result of the {111} orientation can split linear *n*Hex from its branched isomer. To further confirm the overwhelming advantage of {111}-oriented membranes, it is worth trying to synthesize membranes with crystal facets randomly exposed for comparison. We find it possible to slow the growth rate of the {100} crystal facets of HKUST−1 by introducing lauric acid into the support in the SS-TC process[33] and thus obtain near-random-oriented membranes composed of truncated cuboctahedron-shaped crystals (Supplementary Fig. 16). As shown in Supplementary Fig. 17, the intensity ratio (222)/(200) of the membrane is 2.62, close to the value determined based on the calculated powder diffractions (0.96) but much lower than that of the {111}-oriented membrane composed of octahedron-shaped crystals (30.3). The separation performances of these near-random-oriented HKUST-1 membranes were also evaluated. Considering the difference in thickness between {111}-oriented and near-random-oriented membranes, we employed permeability by normalizing the membrane thickness as the permeation metric of membranes for comparison. The near-random-oriented HKUST-1 membranes show $5.31 \times 10^{-14}$ mol m$^{-1}$ s$^{-1}$ Pa$^{-1}$ for the average permeability of *n*Hex but only 8.6 for the selectivity of *n*Hex/23DMB, which confirms the accurate molecular discrimination of triangular-pores running through {111} crystal facets (Fig. 4b). Figure 4c shows a binary component separation chart for C$_5$-C$_7$ linear alkanes over isomers. For example, for the binary component feed, 10/90 *n*Hex/*c*Hex, the average permeance of *n*Hex is $4.44 \times 10^{-7}$ mol m$^{-2}$ s$^{-1}$ Pa$^{-1}$ and the average selectivity of *n*Hex over *c*Hex is 56.9, suggesting the great potential of HKUST-1 membranes for the extraction of low-concentration linear alkanes from isomers including mono-branched, di-branched, and cyclic alkanes that are involved in light naphtha.

## Entropy-driven molecular configuration discrimination

Single-component permeation of alkanes with different molecular kinetic diameters (MKDs) can help to gain a deep insight into the molecular sieving properties of the triangular pores of HKUST-1 membranes, as shown in Fig. 4d. Molecular geometry factors (MGFs) are employed to describe the shapes of alkanes, as shown in Supplementary Table 1. Triangular pores with the dimension of 0.46 nm set a clear permeation cut-off between linear *n*Hex and isomers, which can be translated into the diffusion entropic selectivity immediately reflecting the size/shape discrimination ability of triangular-pore arrays of HKUST-1 membranes (Supplementary Figs. 18–23 and Tables 2–4). Apparently, the diffusion entropic selectivity for *n*Hex over isomers overwhelms other enthalpy-driven adsorption and diffusion factors, and determines the pervaporation separation selectivity for linear alkanes over isomers (Supplementary Tables 2–4). Furthermore, the diffusion entropic selectivity with respect to molecular configuration reaches a high value for *n*Hex over *c*Hex, followed by that for *n*Hex over 23DMB, because of their significant relative MKD and MGF differences, which also reflect the rigidity of triangular-pore arrays of membranes (Fig. 4e). Another phenomenon is that the separation selectivity of the membrane for equimolar *n*Hex/isomer is larger than the ideal permeation selectivity because of the crowding-out effect of the sieving pore.

## Energy-consumption saving and chemical-complexity handling

A detailed chemical process simulation was employed in our study to evaluate the energy consumption of the current technique (Fig. 4f and Supplementary Figs. 24 and 25). Take the separation of the 10/90 *n*Hex/23DMB feed solution as a demonstration. The energy consumption for upgrading *n*Hex (from 10 wt.% to 99 wt.%) and 23DMB (from 90 wt.% to 99 wt.%) by pervaporation through our membrane is estimated to be 0.221 GJ per ton of the feed solution, accounting for only ~9% of the energy consumption for distillation. Compared with MOF adsorbents that are capable of selectively adsorbing linear alkanes but suffer from desorption under necessary high-temperature or vacuum conditions[4], the advantage in the use of our membranes is self-explanatory.

The membrane stability was surveyed with continuous pervaporation. A three-stage separation program involving binary *n*Hex/2MP, *n*Hex/23DMB and ternary *n*Hex/3MP/23DMB was introduced (Fig. 4g). Our membrane is robust; there is no sign of component leakage or pore blocking during pervaporation lasting for over 70 h. The membrane can also maintain stability in an atmospheric environment. When kept in an atmospheric environment for 260 days, the membrane shows that the permeance of *n*Hex and selectivity of *n*Hex/23DMB are almost unchanged (Supplementary Fig. 26). Furthermore, another level of complexity was brought for the feed to simulate real-world separation. Triangular-pore arrayed HKUST-1 membranes can maintain adequate transport properties upon exposure to the ten-component feed. Linear C$_5$-C$_7$ alkanes show appreciably larger permeances through membranes than their mono-branched, bi-branched and cyclical counterparts (Fig. 4h), accounting for 83.6 wt.% of the permeant (Fig. 4i), which not only highlights the size/shape discrimination capability of the triangular-pore array but also demonstrates the potential of our membranes to cope with complexity in real-world mixtures.

In summary, a rigid triangular pore array capable of size/shape discrimination is meticulously structured in an HKUST-1 membrane via an SS-TC process, which enables the efficient separation of linear C$_5$-C$_7$ alkanes from various isomers. Our membranes can function well when they are exposed to a diverse array of alkane isomers, which inspires a deeper exploration using more complex, real-world feed mixtures. The convenience of SS-TC paves the way for synthesizing membranes in a high-throughput and low dose manner that would signify time-savings and cost-efficiency. The universality of the SS-TC method is proven. For example, this process can take place on a TiO$_2$ (5 nm pores) support. The SS-TC method also provides a feasible way to selectively control the growth rate of the specific crystal facet by designing a micro-reaction at the support interface. It can also be used to synthesize near-random-oriented HKUST-1 membranes (Supplementary Figs. 16 and 17) and {100}-oriented ZIF-8 membranes (Supplementary Fig. 27).

# Methods

## Chemicals

Copper(II) nitrate trihydrate [Cu(NO$_3$)$_2$·3H$_2$O, ≥98%], zinc acetate dihydrate [Zn(CH$_3$COO)$_2$·2H$_2$O, ≥98%], BTCA (C$_9$H$_6$O$_6$, 99%), 2-methylimidazole (C$_4$H$_6$N$_2$, 98%), lauric acid [CH$_3$(CH$_2$)$_{10}$COOH, ≥98%] and 23DMB (C$_6$H$_{14}$, 98%) were purchased from Sigma-Aldrich Inc. 22DMB (C$_6$H$_{14}$, 97%), 2MP (C$_6$H$_{14}$, 99%), 3MP (C$_6$H$_{14}$, 99%), *c*Hex (C$_6$H$_{12}$, 99%), 2MB (C$_5$H$_{12}$, 99%), 2MH (C$_7$H$_{16}$, 99%) and 23DMP (C$_7$H$_{16}$, 97%) were purchased from Shanghai Aladdin Biochemical Technology Co., Ltd. *N*,*N*′-dimethylformamide (DMF, C$_3$H$_7$NO, ≥99.5%), *n*Pen, *n*Hex and *n*Hep were supplied by Sinopharm Chemical Reagent Co., Ltd. Ultrapure water was supplied by Hangzhou Wahaha Group Co., Ltd. The 1-mm-thick asymmetric γ-Al$_2$O$_3$ (5 nm pores) and TiO$_2$ (5 nm pores) disks were commercially supplied by Inocermic GmbH (Germany).

## Synthesis of HKUST-1 membranes by the SS-TC method

Metal ion and linker solutions were prepared by dissolving 0.218 g of Cu(NO$_3$)$_2$·3H$_2$O and 0.126 g of BTCA in 10 mL of DMF, respectively. The metal ion solution was slowly added dropwise into the linker solution to form a clear precursor solution. Then the support was placed horizontally in the precursor solution, followed by (a) ultrasonication for 2 min and (b) keeping static at 25 °C for 60 min to fulfil the static seeding process. After that, the alumina support was removed from the precursor solution, and thermal conversion was implemented at 150 °C for 120 min. After an SS-TC cycle, the membranes were

thoroughly washed with DMF and then dried at 120 °C for 12 h under vacuum.

## Synthesis of near-random-oriented HKUST-1 membranes by the SS-TC method
A lauric acid-butanol solution with a concentration of 0.1 mol L$^{-1}$ was prepared and used to thoroughly soak the supports for 1 h. The supports were dried in a vacuum oven at 60 °C for 12 h, followed by the SS-TC cycle mentioned above.

## Synthesis of HKUST-1 powder
A precursor solution was prepared by dissolving 1.82 g of Cu(NO$_3$)$_2$·3H$_2$O and 1.05 g of BTCA in 50 mL of DMF, sealed in a Teflon-lined stainless-steel autoclave and heated at 120 °C in an oven for 24 h, and then cooled naturally to room temperature. The product was washed with DMF and ethanol 3 times and dried at 80 °C for 12 h in a vacuum oven.

## Synthesis of ZIF-8 membranes by the SS-TC method
Metal ion and linker solutions were prepared by dissolving 0.061 g of Zn(CH$_3$COO)$_2$·2H$_2$O and 1.37 g of 2-methylimidazole in 10 mL of H$_2$O, respectively. The metal ion solution was slowly added dropwise into the linker solution to form the precursor solution. Then the support was placed horizontally in the precursor solution, followed by (a) ultrasonication for 10 s and (b) keeping static at 25 °C for 20 min to fulfil the static seeding process. After that, the alumina support was removed from the precursor solution, and thermal conversion was implemented at 60 °C for 120 min. After an SS-TC cycle, the membranes were thoroughly washed with H$_2$O and then dried at 80 °C for 12 h under vacuum.

## Characterizations
XRD patterns of the samples were recorded on a Rigaku instrument (Smartlab) using Cu Kα radiation ($\lambda$ = 0.154 nm at 40 kV and 200 mA). The scan speed was 10° min$^{-1}$, and the 2θ range was 5–50°. The cross-sectional morphology of the membranes was probed by SEM (Quanta 200 FEG, FEI Co.) with an acceleration voltage of 20 kV. The alumina-supported membrane was broken and pasted on a home-made metal holder. Sputter coating of conductive gold nanoparticles on the sample was needed to prevent charge accumulation. EDS elemental mapping of the sample was conducted under a JSM-7800 SEM (Japan electronics Co., LTD). FTIR spectra in the mode of attenuated total reflection were obtained on a Thermo-Fisher spectrometer (Nicolet is50) from 400 to 4000 cm$^{-1}$. Vapor adsorption isotherms of ethane, propane, nHex, 2MP, 23DMB and cHex were obtained at 30 °C and 40 °C via the Hiden Intelligent Gravimetric Analyzer (IGA-100). Raman spectra were collected on a Bruker Optics Senterra Raman spectrometer using a laser with a wavelength of 1064 nm. The size distribution of colloidal particles in the precursor solution was measured using a Zetasizer Nano instrument.

## Pervaporation experiments
Pervaporation through the membrane was conducted at 30 °C. The membrane was sealed into a home-designed stainless-steel module, with an effective membrane area of 0.9499 cm$^2$. Generally, the front side of the membrane was immersed in approximately 50 ml of the feed solution, and the back side was evacuated by an oil pump (Edwards RV12) to maintain a vacuum. When the pervaporation reached a steady state, the permeant was collected in a liquid nitrogen cold trap with an interval of 1–6 h. The composition of the permeant was analyzed by a gas chromatography (GC, Agilent 7890B). The permeation flux ($J$) was calculated via:

$$J = \frac{W}{A \times t} \tag{1}$$

where $W$ is the amount of permeant (kg), $A$ is the effective membrane area (m$^2$), and $t$ is the permeation time (h). The permeance of each component was obtained via:

$$P_i = \frac{J}{(p_f - p_p) \times M_r} \tag{2}$$

where $p_f$ (kPa) and $p_p$ (kPa) are the vapor pressures of component i on the feed and permeate sides, respectively. The $p_f$ of component i was calculated by using Antoine coefficients. The $p_p$ is considered to be zero in the vacuum mode of pervaporation. $M_r$ (g mol$^{-1}$) stands for the molar weight of component i. The permeability of each component was obtained via:

$$P_i \times L = \frac{J \times L}{(p_f - p_p) \times M_r} \tag{3}$$

where $L$ is the membrane thickness. For example, the selectivity ($\alpha$) of nHex over 23DMB was determined by the permeance ratio of nHex to 23DMB:

$$\alpha_{n\text{Hex}/23\text{DMB}} = \frac{P_{n\text{Hex}}}{P_{23\text{DMB}}} \tag{4}$$

## Calculation of diffusion entropic selectivity
Permeability can be expressed as the product of the diffusion coefficient $D_i$ and adsorption coefficient $S_i$:

$$P_i = D_i \times S_i \tag{5}$$

Therefore, the membrane selectivity can be rewritten as the diffusion selectivity product adsorption selectivity:

$$\alpha_{AB} = \frac{P_A}{P_B} = \left(\frac{D_A}{D_B}\right) \times \left(\frac{S_A}{S_B}\right) \tag{6}$$

The adsorption coefficient of each component ($S_i$) equals the quantity adsorbed ($q$) divided by the pressure ($p$). Considering that the adsorption of alkanes on HKUST-1 crystals can be fitted with the Langmuir–Freundlich model (Supplementary Figs. 18–23), $S_i$ is described as:

$$S_i = \frac{q}{p} = \frac{a \times b \times p^{-c}}{1 + b \times p^{1-c}} \tag{7}$$

Based on single-component adsorption isotherms, we can calculate the adsorption selectivity of nHex over its isomers by the ratio of adsorption coefficients between nHex and its isomers (Supplementary Table 3), and further obtain diffusion selectivity according to Eq. (6). Diffusion selectivity is determined by entropic selectivity and enthalpic selectivity[40], namely:

$$\frac{D_A}{D_B} = \left[\exp\left(\frac{S_{D,A} - S_{D,B}}{R}\right)\right]\left[\exp\left(-\frac{E_{D,A} - E_{D,B}}{RT}\right)\right] \tag{8}$$

$$\text{Entropic selectivity} = \exp\left(\frac{S_{D,A} - S_{D,B}}{R}\right) \tag{9}$$

$$\text{Enthalpic selectivity} = \exp\left(-\frac{E_{D,A} - E_{D,B}}{RT}\right) \tag{10}$$

where $R$ is the universal gas constant and $T$ is the absolute temperature. $E_{D,A}$ and $E_{D,B}$ are the diffusion activation energies for guest

molecules A and B, respectively. $S_{D,A}$ and $S_{D,B}$ are the diffusion activation entropies for guest molecules A and B, respectively. The logarithm format of Eq. (8) is:

$$\ln \frac{D_A}{D_B} = \left( -\frac{E_{D,A} - E_{D,B}}{R} \right) \frac{1}{T} + \left( \frac{S_{D,A} - S_{D,B}}{R} \right) \quad (11)$$

Through temperature-variable diffusion selectivity, we can obtain diffusion enthalpic selectivity and diffusion entropic selectivity (Supplementary Table 4).

### Chemical process simulation for *n*Hex/23DMB separation through membrane pervaporation and distillation

Aspen Plus v7.2 software was employed to implement a chemical process simulation for 10/90 *n*Hex/23DMB separation with an annual capacity of 10,000 tons (raw feed solution) to meet the purity target of 99 wt.% for both *n*Hex and 23DMB. The simulated industrial roadmaps for *n*Hex/23DMB separation through membrane pervaporation are shown in Supplementary Figs. 24 and 25, respectively.

## Data availability

The data that support the findings of this study are available within the article and its Supplementary Information. Source data are provided with this paper.

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

## Acknowledgements
This work was supported by the National Natural Science Foundation of China under grant Nos. 22090063 (to W.Y.) and 21978283 (to Y.B.). Y.B. also thanks the Youth Innovation Promotion Association of Chinese Academy of Sciences (2021179).

## Author contributions
Y.B. and W.Y. conceived and supervised the project. Y.B. and Y.W. planned the experiments. Y.W. prepared, characterized, and tested membranes. Z.H. conducted the chemical process simulation. Y.B. and Y.W. analyzed the results and wrote the manuscript. W.Y. reviewed the manuscript.

## Competing interests
The authors declare no competing interests.
