## [Peer Review File · Nature Communications]

REVIEWER COMMENTS

Reviewer #1 (Remarks to the Author):

The manuscript "Energy-efficient extraction of linear alkanes from various isomers using well-structured metal-organic framework membrane " is an interesting and valuable study of fabricating oriented MOF membranes for gas separations. The authors provide a new strategy of separating seeding and growing processes to efficiently synthesize continuous and oriented MOF membrane. The successful extraction of linear alkane from up to ten-component mixtures is impressive and of great importance to practical separation. This work is of great importance and general interest for the readers of Nat Comm, so I would recommend the manuscript for publication. But some questions arise from reading the manuscript:

1. The authors are claiming two advantages of the SS-TC method proposed in this manuscript: first, fabricating continuous membrane; second, growing oriented octahedron dominant membrane. Regarding the second advantage, it is not clear how the splitting of seeding and growing in this method helps control the orientation of grown crystals over other different morphologies. The morphology of the particles in the comparative study (Fig S7) is still mostly octahedron. These facts suggested that it's just a coincidence that the favored facet in crystal growing is the desired one for separation, so I doubt the applicability of this work in other crystal systems when the easily grown facet is not desired for high separation efficiency.
2. The authors characterized the seeds particle (size, IR, Raman) in the solution, but the particles grown in the substrate are not necessarily the same as those in solutions. It is important to understand the morphological information and particle distributions of the seeds layer on the support. The penetration distance of copper seeds into the substrate with respect to seeding time might also be determined.
3. From Fig 4b, the selectivity dropped significantly after the facets are random. What is the ratio of (222) and (200) in this "random" sample? From the PXRD of Fig S7, (222) seems still the dominant facet, which does not support their claim of random distribution.
4. The comparisons of oriented/random faceted membranes should be either both pure MOFs membranes or both MMMs, instead of one MOF membrane and one MMM, which is misleading and hard to draw a conclusion.
5. What is the practical composition of linear alkane in the industrial mixture to separate? Is it 10%?
6. Did this method enhance the binding of MOF particles with the substrate stronger than other methods such as those obtained from the regular solvothermal method?
7. Why did the selectivity increase significantly in 2 cycle SS-TC sample? Is that because of cracking or a high content of other facets in 1 and 3 cycle SS-TC samples? The PXRD should also be provided for all samples.
8. What is the BET surface area of the HKUST-1 powder?
9. I wonder why the authors use MGF to quantify the difference of the molecules. The difference of ethane/propane over heptane is very large but they can all permeate through the pore, so the MGF difference does not always have direct relation with the selectivity. Instead, the molecules are passing through the pore window using their smallest cross sections, so it might be a better parameter.
10. Modeling study of the diffusion of linear and branched alkanes through different pore windows will help understand the mechanisms.

Reviewer #2 (Remarks to the Author):

Separation of alkanes (linear, branched, cyclic/aromatic) is an crucial task in chemical industry for their various end uses. This has been extensively explored by column separation using zeolites and MOFs. In contrast, separation of alkanes by membranes has not been well explored, while membrane separation of light gases (H₂, CO₂, C₂/C₃ etc) attract more scientific attention. In the current work, the authors successfully prepared "(222)-oriented HKUST-1 membranes" by controlling the processes of "static seeding," "thermal conversion" and "SS-TC cycles." The membrane exhibited encouraging separation performance for alkane isomers. Overall, the study is

well conducted, representing certain advances in membrane fabrication with specific orientation, and is a nice trial of MOF membrane for alkane separations. It could become suitable for publication in Nat. Commun., however, after addressing the following concerns.

1. The formation process and trend of (222) crystal facets were thoroughly studied by controlling the thermal transition time and temperature. However, specific mechanisms or principles behind the formation of oriented (222) crystal facets were not explained. Readers would like to see more related discussions.
2. In the gas separation section, although the authors compared the gas separation performance of random orientation HKUST-1 mixed matrix membranes with that of (222)-oriented HKUST-1 membranes, this may not be convincing enough. It is suggested to prepare randomly oriented HKUST-1 membranes and conduct gas separation performance tests to further demonstrate the significance of crystal facet orientation in alkane separation.
3. The authors should compare and comment on the separation performance of the current membrane and the reported ones, such as the recent report of MOF membrane for alkane separation (Journal of Membrane Science 661 (2022) 120916)
4. Resolution of some of the figures need to be improved. For example, some panels in Fig. 4 and Fig. S8.
5. In Fig. S8 and Table. S2, the adsorption capacity, adsorption selectivity and diffusion selectivity of C3-C7 was tested and calculated, respectively. Please give the intuitive permeation flux, gas adsorption capacity and the corresponding pressure. It is very important for entropic selectivity and enthalpic selectivity. In addition, is the separation selectivity (α) in Table 2 refer to ideal selectivity? Why is the selectivity of nHex/23DMB (11.5) in table S2 significantly lower than the average selectivity of 139? These should be clarified in the manuscript.

Reviewer #3 (Remarks to the Author):

The authors reported oriented HKUST-1 membranes with exposed (222) planes through a low dose synthesis method (static seeding-thermal conversion) for terminating undesired crystal nucleation and growth. The resulted triangular sieving pore array of (222) induces good separation for C5 and C7 with less energy than the vacuum distillation. The synthesis is process is interesting and may scalable for industry. The following comments should be considered.

1. From the XRD, the (111) peak is relatively weak for the SSTC formed HKUST-1 membrane, since it is the same direction of (222). What is the reason? By the way, the orientation typical named as the direction zone $\langle 111 \rangle$, but not use the index for plane.
2. Adding simulation for the separation mechanism might be better for understanding.
3. The static seeding thermal conversion is relatively time consuming. How about the thickness of AAO affect the membrane growth?
4. How about the long-term stability of the resulted membrane in atmospheric environment?
5. Is it possible to extend this process for other substrate?

Reviewer #4 (Remarks to the Author):

High-quality crystal nucleation and growth at the interface is critical for a molecular sieve membrane. Wang and co-workers reported a two-step SS-TC method that is meticulously used to isolate crystal nucleation and growth and balance the competition between them, ultimately producing highly oriented HKUST-1 membranes with fully opened triangular sieving apertures. This is a comprehensive work, and achieves harmony of the architecture design and separation purpose. Another impressive progress in this work is to evaluate the membrane potential by using complex multicomponent mixtures rather than idealized binary mixtures. Only a very small fraction of the literature in the field of chemical separations reports such data. It can be accepted by Nature Communications after these minor remarks can be addressed.

1. Did the membrane layer penetrate into the alumina support in the current experimental setting?
2. I notice ultrasonic was employed at inception of the static seeding. Elucidate its effect on the pre-nucleation and membrane formation.
3. In Fig. S6, I notice the permeance and separation selectivity of the membrane decreased after 3 cycles of SS-TC. I would like to know the orientation index, e.g. intensity ratio (222)/(200) of the membrane after 2 and 3 cycles of SSTC. Is this separation trend related to the change of the orientation?
4. Can you quantify the weight of pre-nucleation solution that was entrapped into the alumina support?
5. My impression is that the TC process was conducted in an open vessel. What's the separation result of the membrane if the TC process was sealed in a closed system? Did the solvent evaporation concurrent with the crystal growth influence the membrane formation?

RESPONSE TO REVIEWERS' COMMENTS

Reviewer #1:

The manuscript "Energy-efficient extraction of linear alkanes from various isomers using well-structured metal-organic framework membrane" is an interesting and valuable study of fabricating oriented MOF membranes for gas separations. The authors provide a new strategy of separating seeding and growing processes to efficiently synthesize continuous and oriented MOF membrane. The successful extraction of linear alkane from up to ten-component mixtures is impressive and of great importance to practical separation. This work is of great importance and general interest for the readers of Nat Comm, so I would recommend the manuscript for publication. But some questions arise from reading the manuscript:

1. The authors are claiming two advantages of the SS-TC method proposed in this manuscript: first, fabricating continuous membrane; second, growing oriented octahedron dominant membrane. Regarding the second advantage, it is not clear how the splitting of seeding and growing in this method helps control the orientation of grown crystals over other different morphologies. The morphology of the particles in the comparative study (Fig S7) is still mostly octahedron. These facts suggested that it's just a coincidence that the favoured facet in crystal growing is the desired one for separation, so I doubt the applicability of this work in other crystal systems when the easily grown facet is not desired for high separation efficiency.

Thank you for your suggestion. It is known that crystal facets with a slow growth rate generally dominate the morphology of the crystal. In our study, we found that crystals growing in the bulk solution reveal a random shape, typified by a truncated cuboctahedron shape with a significant proportion of {100} facets. We supplemented Fig. S2 to illustrate a comparative study. From these figures, we can easily tell the morphology difference between crystals growing at the support interface (sharp octahedron) and crystals growing at the bulk liquid phase (truncated cuboctahedron). A brief conclusion is that the difference in the growth rate between the {100} and {111} crystal facets can be further widened at the support interface with a sharper vertical gradient in the concentration of nutrients. The SS-TC method involves (1) crystal pre-nucleation at the solution-alumina interface at room temperature and (2) thermal conversion by removing the bulk solution to trigger seed growth in a self-limited

chemical environment that lacks nutrients. Splitting seeding and growing by using this method can help terminate undesired crystal growth in a homogenous bulk solution and confine the crystal growth at the support interface. This is a vital prerequisite for the synthesis of highly oriented HKUST-1 membranes composed of octahedron shaped HKUST-1 building blocks. The detailed explanation was supplemented in the revised manuscript.

In addition, we also demonstrated the wide applicability of the SS-TC method. We supplemented two experimental results. First, by extensive *trial and error*, we found it possible to slow the growth rate of the {100} crystal facet of HKUST-1 by introducing lauric acid into the support in the SS-TC process and thus obtain randomly oriented membranes composed of truncated cuboctahedron shaped crystals (Supplementary Fig. S16 and S17). The intensity ratio (222)/(200) of the membrane is 2.62, close to the value determined based on the calculated powder diffractions but much lower than that of the {111}-oriented membrane composed of octahedron shaped crystals. That is to say, the SS-TC method provides a feasible way to selectively control the growth rate of the specific crystal facet by designing a micro-reaction at the support interface, regardless of whether the crystal facet is easily grown. Second, the SS-TC method can also be used to synthesize {100}-oriented ZIF-8 membranes (Supplementary Fig. S27).

2. The authors characterized the seeds particle (size, IR, Raman) in the solution, but the particles grown in the substrate are not necessarily the same as those in solutions. It is important to understand the morphological information and particle distributions of the seeds layer on the support. The penetration distance of copper seeds into the substrate with respect to seeding time might also be determined.

Thank you for your suggestion. We tried to directly characterize the pre-nucleation seed layer on the alumina support by SEM. Except for scattered crystals, we cannot obtain any surface morphology information of the pre-nucleation seeds because (1) these pre-organized metal-linker species on the top surface of the support might be susceptible to the high-powered electron beam and (2) they might penetrate into the alumina disk. According to your suggestion, we conducted element analysis along the cross-section of the disk by using EDS to probe the distribution of pre-nucleation seeds. A clear copper distribution along the cross-section of the disk is shown in Supplementary Fig. S4. Furthermore, the penetration depth of pre-nucleation seeds slightly changed with the seeding time, ranging from 0.60 to 0.82 μm against the seeding time from 30 to 120 min.

3. From Fig 4b, the selectivity dropped significantly after the facets are random. What is the ratio of (222) and (200) in this “random” sample? From the PXRD of Fig S7, (222) seems still the dominant facet, which does not support their claim of random distribution.

Thank you for your suggestion. According to your comment point 4, we synthesized a pure HKUST-1 membrane with a near-random orientation for comparison. Please see the response to point 4.

4. The comparisons of oriented/random faceted membranes should be either both pure MOFs membranes or both MMMs, instead of one MOF membrane and one MMM, which is misleading and hard to draw a conclusion.

Thank you for your suggestion. According to your comment, we synthesized near-random-oriented pure-phase HKUST-1 membranes composed of truncated cuboctahedron-shaped crystals by introducing lauric acid into the support in the SS-TC process (Supplementary Fig. S16 and S17). As shown in Supplementary Fig. S17, the intensity ratio (222)/(200) of the membrane is 2.62, close to the value determined based on the calculated powder diffractions (0.96) but much lower than that of the {111}-oriented membrane composed of octahedron-shaped crystals. The separation performances of these near-random-oriented HKUST-1 membranes were also evaluated. Considering the difference in the thickness between {111}-oriented and near-random-oriented membranes, we employed permeability by normalizing the membrane thickness as the permeation metric of the membranes for comparison. The near-random-oriented HKUST-1 membranes show $5.31 \times 10^{-14} \text{ mol m}^{-1} \text{ s}^{-1} \text{ Pa}^{-1}$ for the average permeability of *n*Hex but only 8.6 for the selectivity of *n*Hex/23DMB, which confirms the accurate molecular discrimination of triangular-pores running through {111} crystal facets (as shown in Fig. 4b). The above results and discussion were supplemented in the revised manuscript.

5. What is the practical composition of linear alkane in the industrial mixture to separate? Is it 10%? **Yes. It has been reported that the linear *normal*-Pentane, *normal*-Hexane and *normal*-Heptane are approximately 4-10 wt.% in light naphtha (G. W. Meindersma, *Extraction of aromatics from naphtha with ionic liquids: from solvent development to pilot RDC evaluation, 2005*).**

6. Did this method enhance the binding of MOF particles with the substrate stronger than other methods such as those obtained from the regular solvothermal method?

Yes. The HKUST-1 membrane derived from the SS-TC method is closely connected to the support, similar to the MOF membrane synthesized by direct reaction of the alumina support with a linker (*Angew. Chem. Int. Ed.* 2022, 61, e202114479; *Angew. Chem. Int. Ed.* 2023, 62, e202302181). After eight ultrasonic treatments (30 min for each time), the permeance of *n*Hex and selectivity of *n*Hex/23DMB for the 10/90 *n*Hex/23DMB feed solution maintain constant (Supplementary Fig. S15).

7. Why did the selectivity increase significantly in 2 cycle SS-TC sample? Is that because of cracking or a high content of other facets in 1 and 3 cycle SS-TC samples? The PXRD should also be provided for all samples.

Thank you for your suggestion. We supplemented the analysis of the orientation and thickness of membranes, as shown in Fig. S9 and S10. After 2 cycles of SS-TC, the thickness of the membrane was increased to 1.7 μm , whereas the intensity ratio (222)/(200) was barely changed, concurrent with a significant improvement in the separation selectivity for *n*Hex/23DMB (Fig. S11). We speculated that 2 cycles of SS-TC can eliminate minor local defects in the membranes completely, thus leading to a significant improvement in the isomer separation accuracy. After 3 cycles of SS-TC, the intensity ratio (222)/(200) decreased to 18.6, suggesting that the proportion of {100} crystal facets aligned out-of-plane was not negligible. The molecular sieving property was thus softened, corresponding to a reduction in the selectivity for *n*Hex/23DMB. These results and explanations were supplemented in the revised manuscript.

8. What is the BET surface area of the HKUST-1 powder?

The BET surface area of the HKUST-1 powder is 1244 $\text{m}^2 \text{g}^{-1}$ according to the experimental N_2 adsorption isotherm at 77 K (Supplementary Fig. S1). This result was supplemented in the revised manuscript.

9. I wonder why the authors use MGF to quantify the difference of the molecules. The difference of ethane/propane over heptane is very large but they can all permeate through the pore, so the MGF difference does not always have direct relation with the selectivity. Instead, the molecules are passing through the pore window using their smallest cross sections, so it might be a better parameter.

Thank you for your suggestion. The permeation behavior of molecules through the sieving pores of the membrane is widely accepted to correlate with the molecular kinetic diameter

(MKD) of the permeant. Therefore, as you suggested, we employed MKD as a main measure to illustrate the single-component permeation of alkanes, as shown in Fig. 4d. In addition to MKD, MGF is also an important and commonly used shape descriptor for alkane isomers, which is thought to be a contributor to the permeation selectivity especially to diffusion entropic selectivity (*Science* 2013, 340, 960-964). Therefore, when we discussed the diffusion entropic selectivity between *n*Hex and its isomer (Fig. 4e), we also employed MGF to help readers to understand the molecular shape.

10. Modeling study of the diffusion of linear and branched alkanes through different pore windows will help understand the mechanisms.

Thank you for your suggestion. The separation of linear and branched alkanes through our HKUST-1 membrane follows the molecular sieving principle. This point was illustrated reasonably well by comparing the separation properties between {111}-oriented membranes and near-random-oriented membranes. We agree with you. Modelling is a method to help understand the mechanism. However, simulation of the diffusion of linear and branched alkanes through different pore windows is a very complex computational task. It is very time-consuming and unpredictable especially for the current petrochemical system, which involves up to ten components in our study. We apologize that the corresponding simulation has not yet been achieved in our lab.

Reviewer #2:

Separation of alkanes (linear, branched, cyclic/aromatic) is a crucial task in chemical industry for their various end uses. This has been extensively explored by column separation using zeolites and MOFs. In contrast, separation of alkanes by membranes has not been well explored, while membrane separation of light gases (H₂, CO₂, C₂/C₃ etc) attract more scientific attention. In the current work, the authors successfully prepared “(222)-oriented HKUST-1 membranes” by controlling the processes of “static seeding,” “thermal conversion” and “SS-TC cycles.” The membrane exhibited encouraging separation performance for alkane isomers. Overall, the study is well conducted, representing certain advances in membrane fabrication with specific orientation, and is a nice trial of MOF membrane for alkane separations. It could become suitable for publication in Nat. Commun., however, after addressing the following concerns.

1. The formation process and trend of (222) crystal facets were thoroughly studied by controlling the thermal transition time and temperature. However, specific mechanisms or principles behind the formation of oriented (222) crystal facets were not explained. Readers would like to see more related discussions.

Thank you for your suggestion. The detailed discussion was supplemented in the revised manuscript.

“Here, we also discuss the orientation evolution mechanism with thermal conversion time. The thermal conversion process by removing the bulk solution is thought to be a process in which the crystal growth was completely confined at the support interface. The instantaneous high-temperature environment can prompt the fast growth of seeds on the surface of the support by means of limited nutrients in the support. This self-limiting chemical condition at the support interface can widen the difference in the growth rate between {100} and {111} crystal facets, leading to a membrane composed of octahedral crystals. With the nutrients gradually depleted, the growth rate of {111} crystal facets could be further slowed down, thus leading to an increase in the proportion of triangular {111} facets aligned out-of-plane with time.”

2. In the gas separation section, although the authors compared the gas separation performance of random orientation HKUST-1 mixed matrix membranes with that of (222)-oriented HKUST-1 membranes, this may not be convincing enough. It is suggested to prepare randomly oriented HKUST-1 membranes and conduct gas separation performance tests to further demonstrate the significance of crystal facet orientation in alkane separation.

Thank you for your suggestion. By extensive *trial and error*, we found it possible to slow the growth rate of the {100} crystal facet of HKUST-1 by introducing lauric acid into the support in the SS-TC process and thus obtain compact and near-random-oriented membranes composed of truncated cuboctahedron-shaped crystals (Supplementary Fig. S16 and S17). As shown in Supplementary Fig. S17, the intensity ratio (222)/(200) of the membrane is 2.62, close to the value determined based on the calculated powder diffractions (0.96) but much lower than that of the {111}-oriented membrane composed of octahedron shaped crystals. The separation performances of these near-random-oriented HKUST-1 membranes were also evaluated. Considering the difference in the thickness for {111}-oriented and near-random-oriented membranes, we employed permeability by normalizing the membrane thickness as the permeation metrics of membranes for comparison. The near-random-oriented HKUST-

1 membranes show $5.31 \times 10^{-14} \text{ mol m}^{-1} \text{ s}^{-1} \text{ Pa}^{-1}$ for the average permeability of *n*Hex but only 8.6 for the average selectivity of *n*Hex/23DMB, which confirms the accurate molecular discrimination of triangular-pores running through {111} crystal facets (Fig. 4b). The above results and discussion were supplemented in the revised manuscript.

3. The authors should compare and comment on the separation performance of the current membrane and the reported ones, such as the recent report of MOF membrane for alkane separation (Journal of Membrane Science 661 (2022) 120916).

Thank you for your suggestion. We supplemented the comparison according to your recommendation.

“HKUST-1 membranes exhibit both significant *n*Hex permeance and *n*Hex/23DMB selectivity, far beyond the recently reported Al-bttotb membrane.”

4. Resolution of some of the figures need to be improved. For example, some panels in Fig. 4 and Fig. S8.

Thank you for your suggestion. We split and rearranged the panels of this supplementary figure for better resolution, as shown in Supplementary Fig. S18-S23 in the revised manuscript. Meanwhile, we used high-resolution figures (≥ 600 dpi) in the main text.

5. In Fig. S8 and Table. S2, the adsorption capacity, adsorption selectivity and diffusion selectivity of C3-C7 was tested and calculated, respectively. Please give the intuitive permeation flux, gas adsorption capacity and the corresponding pressure. It is very important for entropic selectivity and enthalpic selectivity. In addition, is the separation selectivity (α) in Table 2 refer to ideal selectivity? Why is the selectivity of *n*Hex/23DMB (11.5) in table S2 significantly lower than the average selectivity of 139? These should be clarified in the manuscript.

Thank you for your suggestion. We supplemented Tables S2 and S3 to list single-component permeances, gas adsorption coefficients and the corresponding vapour pressure. The separation selectivity in Table S3 refers to ideal permeation selectivity. We revised this description to avoid misunderstanding. The average separation selectivity of the membrane was 139 and 51.6 in the case of the 10/90 and 50/50 *n*Hex/23DMB feed solutions, respectively. The separation selectivity was larger than the ideal permeation selectivity because of the crowding-out effect of the sieving pore. This explanation was supplemented in the revised manuscript.

Reviewer #3:

The authors reported oriented HKUST-1 membranes with exposed (222) planes through a low dose synthesis method (static seeding-thermal conversion) for terminating undesired crystal nucleation and growth. The resulted triangular sieving pore array of (222) induces good separation for C5 and C7 with less energy than the vacuum distillation. The synthesis is process is interesting and may scalable for industry. The following comments should be considered.

1. From the XRD, the (111) peak is relatively weak for the SSTC formed HKUST-1 membrane, since it is the same direction of (222). What is the reason?

Thank you for your review. The intensity of the X-ray diffraction is correlated to the structure factor F_{hkl} . Based on the crystallographic CIF file of HKUST-1, the F_{hkl} value for the (222) diffraction is 2170.00, much larger than that for the (111) diffraction (172.26). Therefore, we speculate that the (222) diffraction is intrinsically stronger than the (111) diffraction.

By the way, the orientation typical named as the direction zone $\langle 111 \rangle$, but not use the index for plane.

Thank you for your kind reminder. We revised the corresponding description to “{111}-oriented membrane”.

2. Adding simulation for the separation mechanism might be better for understanding.

Thank you for your suggestion. The separation of linear and branched alkanes through our HKUST-1 membrane follows the molecular sieving principle. This point was illustrated reasonably well by comparing the separation properties between {111}-oriented membranes and near-random-oriented membranes. Modelling is a method to help understand the mechanism. However, simulation of the diffusion of linear and branched alkanes through different pore windows is a very complex computational task. It is very time-consuming and unpredictable especially for the current petrochemical system, which involves up to ten components in our study. We apologize that the corresponding simulation has not yet been achieved in our lab.

3. The static seeding thermal conversion is relatively time consuming. How about the thickness of AAO affect the membrane growth?

In fact, the static seeding-thermal conversion (SS-TC) process is not time consuming. It takes 182 min to complete an SS-TC cycle because there is only an approximate 1 μm -thick-layer growing on the top surface of the support rather than through the support. This point can be proven by XRD. There is no MOF signal on the back side of the support. Therefore, the thickness of the support cannot affect membrane growth.

Fig. 1 XRD patterns of the back side of the HKUST-1 membrane after 2 SS-TC cycles.

4. How about the long-term stability of the resulted membrane in atmospheric environment?

Thank you for your suggestion. According to your comment, we re-examined the separation performance of the membrane prepared in October 2022. When kept in an atmospheric environment for 260 days, the membrane shows that the permeance of *n*Hex and selectivity of *n*Hex/23DMB are almost unchanged (Supplementary Fig. S26). This result was supplemented in the revised manuscript.

5. Is it possible to extend this process for other substrate?

Yes. The universality of this process is proven. A similar microscopic morphology can also be found in the membrane prepared on the TiO₂ support (Supplementary Fig. S13 and S14). After 2 cycles of the SS-TC process, the separation selectivity of the membranes for 10/90 *n*Hex/23DMB is 52. These results were supplemented in the revised manuscript.

Reviewer #4:

High-quality crystal nucleation and growth at the interface is critical for a molecular sieve membrane. Wang and co-workers reported a two-step SS-TC method that is meticulously used to isolate crystal nucleation and growth and balance the competition between them, ultimately producing highly oriented HKUST-1 membranes with fully opened triangular sieving apertures. This is a comprehensive work, and achieves harmony of the architecture design and separation purpose. Another impressive progress in this work is to evaluate the membrane potential by using complex multicomponent mixtures rather than idealized binary mixtures. Only a very small fraction of the literature in the field of chemical separations reports such data. It can be accepted by Nature Communications after these minor remarks can be addressed.

1. Did the membrane layer penetrate into the alumina support in the current experimental setting?

Thank you for your review. Yes. As shown in Supplementary Fig. S5, there is an approximately 0.3- μm -thick layer confined into the alumina support after 1 SS-TC cycle, which is proven by the penetration of copper into the Al-enrichment layer.

2. I notice ultrasonic was employed at inception of the static seeding. Elucidate its effect on the pre-nucleation and membrane formation.

Thank you for your suggestion. We supplemented Fig. S2 to show the top-view SEM image of the HKUST-1 membrane without ultrasonication before static seeding. This is positive evidence to indicate that ultrasonication can help overcome the limitation of the triple phase boundary involving alumina, air (in the pores of alumina) and liquid, expel air in the pores of the alumina support and make nutrients completely wet the support. Without ultrasonication, there would be a lot of hole defects in the membrane layer and between the membrane layer and the support because of the air escape from the pores of the support during membrane preparation. This result and discussion were supplemented in the revised manuscript.

3. In Fig. S6, I notice the permeance and separation selectivity of the membrane decreased after 3 cycles of SS-TC. I would like to know the orientation index, e.g. intensity ratio (222)/(200) of the membrane after 2 and 3 cycles of SSTC. Is this separation trend related to the change of the orientation?

Thank you for your review. Yes. After 3 cycles of SS-TC, the intensity ratio (222)/(200) decreased to 18.6, suggesting that the proportion of {100} crystal facets aligned out-of-plane

was not negligible. The molecular sieving property was thus softened, corresponding to a reduction in the selectivity for *n*Hex/23DMB. According to your suggestion, these results and explanations were supplemented in the revised manuscript, as shown in Figs. S9-S11.

4. Can you quantify the weight of pre-nucleation solution that was entrapped into the alumina support?

Thank you for your review. The weight of the alumina disk increased by 21% because of seeding, together with the capillary adsorption of nutrients on the surface of the support. This result was also mentioned in the main text.

5. My impression is that the TC process was conducted in an open vessel. What's the separation result of the membrane if the TC process was sealed in a closed system? Did the solvent evaporation concurrent with the crystal growth influence the membrane formation?

Thank you for your review. According to your comment, we supplemented the experiments. We found that a similar microscopic morphology, orientation and isomer separation performance can be achieved in the membrane if the thermal conversion process is sealed in a closed system (Supplementary Fig. S12). The selectivity of the HKUST-1 membrane for the 10/90 *n*Hex/23DMB feed solution is 122, which is comparable to that of the membrane prepared in an open vessel. Solvent evaporation has a negligible effect on the resultant membrane.

REVIEWERS' COMMENTS

Reviewer #1 (Remarks to the Author):

The authors have addressed the reviewers' comments properly. The manuscript is recommended for acceptance now.

Reviewer #2 (Remarks to the Author):

The authors have made appropriate revisions and my concerns have been addressed. Recommend to publish as is.

Reviewer #3 (Remarks to the Author):

All the comments have been well addressed. The current form is acceptable.

Reviewer #4 (Remarks to the Author):

I would recommend the manuscript for publication.

RESPONSE TO REVIEWERS' COMMENTS

Reviewer #1:

The authors have addressed the reviewers' comments properly. The manuscript is recommended for acceptance now.

Thank you for your review.

Reviewer #2:

The authors have made appropriate revisions and my concerns have been addressed. Recommend to publish as is.

Thank you for your review.

Reviewer #3:

All the comments have been well addressed. The current form is acceptable.

Thank you for your review.

Reviewer #4:

I would recommend the manuscript for publication.

Thank you for your review.